# Investigating the prevalence of cognitive impairment and dementia in the Northern Ireland Cohort for the Longitudinal Study of Ageing (NICOLA): the Harmonised Cognitive Assessment Protocol (HCAP) cross-sectional substudy

Leeanne O'Hara ,[1] Charlotte Neville,[1] Calum Marr,[1] Michael McAlinden,[1] Frank Kee,[1] David Weir,[2] Bernadette McGuinness[1]

¹Centre for Public Health, Queen's University Belfast, Belfast, UK
²University of Michigan, Ann Arbor, Michigan, USA

**Correspondence to**
Dr Leeanne O'Hara;
l.ohara@qub.ac.uk

## ABSTRACT

**Introduction** The Northern Ireland Cohort for the Longitudinal Study of Ageing (NICOLA) study is the largest study of ageing in Northern Ireland (NI). The Harmonised Cognitive Assessment Protocol (HCAP) is a substudy of NICOLA designed to assess cognitive impairment and dementia in individuals aged 65 and over. The NICOLA-HCAP substudy is funded by the National Institute on Aging as part of a network for enhancing cross-national research within a worldwide group of population-based, longitudinal studies of ageing, all of which are centred around the US-based Health and Retirement Study.

**Methods and analysis** The NICOLA-HCAP study will draw on the main NICOLA cohort (of 8283 participants) and randomly sample 1000 participants aged 65 and over to take part in the substudy. Participants will complete a series of cognitive tests (n=19) via a computer-assisted personal interview administered in their home (or alternatively within the research centre) and will be asked to nominate a family member or friend to complete an additional interview of validated instruments to provide information on respondent's prior and current cognitive and physical functioning and whether the individual requires help with daily activities. The objectives of the study are: to investigate the prevalence of dementia and cognitive impairment in NICOLA; harmonise scoring of the NICOLA-HCAP data to the HCAP studies conducted in Ireland, the USA and England; to explore the validity of dementia estimates; and investigate the risk factors for dementia and cognitive impairment.

**Ethics and dissemination** The study received ethical approval from the Faculty of Medicine, Health and Life Sciences Research Ethics Committee, Queen's University Belfast. We will provide data from the Northern Irish HCAP to the research community via data repositories such as the Dementias Platform UK and Gateway to Global Aging to complement existing public data resources and support epidemiological research by others. Findings will also be disseminated through peer-reviewed publications and at international conferences.

## STRENGTHS AND LIMITATIONS OF THIS STUDY

⇒ The strength of Northern Ireland Cohort for the Longitudinal Study of Ageing-Harmonised Cognitive Assessment Protocol (NICOLA-HCAP) is that the substudy administers the same measures as the Health and Retirement Study (HRS; USA), The Irish Longitudinal Study on Ageing and the English Longitudinal Study of Ageing (ELSA)-HCAP studies, and is part of a wider network of HCAP studies (including, for example, Mexico, India, China and Europe). Further, the measures are all objective measures of cognitive health.

⇒ The scope and detail of data collected within the NICOLA study allow for in-depth analysis and understanding of factors that affect cognition.

⇒ A limitation to the work is that NICOLA-HCAP is currently a cross-sectional study. The aim is to conduct further follow-up studies as per HRS and ELSA.

⇒ As with any cohort substudy there is also a risk of respondent burden. NICOLA Wave 3, which is due to commence at the end of 2023, will also include measures of cognition. It will therefore be necessary to stage data collection with those who participated in the NICOLA-HCAP substudy to reduce burden and minimise the risk of respondent familiarity with certain tests.

## INTRODUCTION

### The Northern Ireland Cohort for the Longitudinal Study of Ageing

The Northern Ireland Cohort for the Longitudinal Study of Ageing (NICOLA) is the largest public health cohort study of ageing in Northern Ireland (NI).[1] In brief, NICOLA was designed to maximise comparability with other well-established longitudinal studies

of ageing, including the Health and Retirement Study (HRS)[2] in the USA, The Irish Longitudinal Study on Ageing (TILDA)[3] and the English Longitudinal Study of Ageing (ELSA).[4] The study began in 2013 by recruiting a representative sample of 8283 individuals aged 50 years and over living across NI. The first wave included three main components: a computer-assisted personal interview (CAPI) conducted in the participant's own home; a self-completion questionnaire including a dietary questionnaire; and an objective health assessment. Wave 2 commenced in 2017 and consisted of a follow-up CAPI and self-completion questionnaire. A refresh of the NICOLA cohort sample also took place at the end of Wave 2 to enable the maintenance and representation of new participants aged 50–54 years. A further add-on to Wave 2 was the mailing of a self-completion COVID-19 questionnaire to all study participants in February 2021.[5] It is hoped that Wave 3 will commence later in 2023 and it is the intention that the cohort will continue to be followed up for a period of at least 10 years, pending continued funding. The NICOLA study will enable researchers to gain a better understanding of factors that contribute to health and social outcomes in the older population in NI.[6]

### The NICOLA Harmonised Cognitive Assessment Protocol
The Harmonised Cognitive Assessment Protocol (HCAP) is a substudy of NICOLA designed to assess dementia and cognitive impairment in a sample of NICOLA participants aged 65 years and older. Funded by the National Institute on Aging, the NICOLA-HCAP study is part of a framework for enhancing cross-national research within a group of studies worldwide, all of which are centred on the US-based HRS.[2] The NICOLA-HCAP study is being carried out at the Centre for Public Health at Queen's University, Belfast. This paper describes the cohort profile HCAP and the findings from the pilot study.

### Justification for the HCAP substudy
#### Dementia and mild cognitive impairment
Dementia is a growing public health concern that currently affects more than 55 million people across the world. There are approximately 10 million new cases diagnosed across the globe annually and it is estimated that by 2050, a total of 152 million people will be living with dementia worldwide.[6] In the UK, dementia is the third most common cause of death for men and the leading cause of death among women.[7]

Dementia is a syndrome characterised by the progressive and irreversible deterioration of cognitive functioning that affects an individual's thinking, remembering and reasoning as well as behavioural abilities.[8 9] Dementia is a result of neurodegeneration caused by diseases, such as Alzheimer's disease (AD), vascular dementia, dementia with Lewy bodies and frontotemporal dementia or a series of strokes. While many cases reflect a mixture of pathologies, AD is the most common cause of dementia,

accounting for approximately 60–70% of dementia diagnoses.[10]

Cognitive impairment is characterised by difficulties in remembering previously acquired knowledge, learning new things, concentrating or making decisions that affect everyday life.[11] Cognitive impairment ranges from mild to severe. People with mild impairment may begin to notice changes in cognitive functions, but still be able to carry out everyday activities. However, severe levels of impairment can lead to loss of the ability to understand the meaning or importance of something and the ability to talk or write, resulting in the inability to live independently. Mild cognitive impairment (MCI) is a condition in which someone has minor problems with their cognitive abilities such as memory or thinking.[12] In MCI, these difficulties are worse than would normally be expected for a healthy person of their age.[13] A person with MCI is more likely to go on to develop dementia at a rate of 10–15% each year[14] and as the 'Baby Boomer' generation pass the age of 65, the number of people living with MCI is expected to jump dramatically.[15]

### Risk factors for dementia and MCI
There is growing evidence that a large proportion of dementia cases might be delayed or prevented with effective control of modifiable risk factors. While ageing is the strongest known risk factor for dementia, it does not exclusively affect older people, with young-onset dementia representing 9% of cases.[6] Other risk factors for dementia include lifestyle factors such as sedentary behaviour,[16] smoking,[17] excess alcohol consumption,[18] being overweight,[19] poor diet,[20] high cholesterol,[21] high blood sugar levels[22] and uncontrolled blood pressure.[23] Hearing loss, depression, traumatic brain injury and pollution are also noted as potentially modifiable risk factors.[24]

Further, an array of sociodemographic factors is associated with an increased risk of dementia including living in areas of socioeconomic deprivation[25 26] and areas with low educational attainment.[27] Loneliness is linked to an increased risk of dementia,[28] and social isolation has been shown to increase the risk of dementia by 50%.[29] The impact of both of these factors has been exacerbated by the COVID-19 pandemic.[30] The consensus among international experts at the G8 Dementia Summit in 2013 was that increasing physical activity and fruit and vegetable consumption, controlling weight, diabetes and high blood pressure, smoking cessation and avoiding excessive alcohol intake could prevent a substantial proportion of dementia cases.[31]

### Prevalence and cost of dementia
The current estimates of dementia prevalence in the UK indicate that around 7.1%, representing 1 in every 14 of the population aged 65 years and older, are diagnosed with dementia.[32] Similarly, in North America, 10% of individuals aged 65 and over receive a dementia diagnosis.[33] However, a substantial number of cases are not diagnosed, leading to difficulties in estimating true

prevalence. There is a pressing need for high-quality data from nationally representative samples of older men and women, hence the need for the HCAP study.[33] Reliable national data on incidence and prevalence of dementia and cognitive impairment are vital for service planning, the prediction of future needs, estimating the costs of dementia care and understanding the impact of these conditions on individuals and their families. There is also a strong research need for robust estimates of MCI and dementia in order to facilitate studies of the aetiology and consequences of these conditions.

Although the number of people with dementia is projected to increase throughout the world because of a well-documented demographic shift towards rising numbers of older adults,[34] estimates of future levels are complicated by the assumptions of the underlying methods used to derive them. There is evidence from both England and the USA that the incidence of dementia has declined over the past decades,[35] but it is rising exponentially in low and middle-income countries.[36]

The marked increase in numbers of individuals expected to have dementia has important economic implications. With an estimate of US$13 trillion annually attributed to the condition, an increase in prevalence will have significant economic impact globally.[37] The corresponding economic cost to the UK of caring for people with dementia is estimated to grow from £34.7 billion (2019) to £94.1 billion by 2040.[38] As the population ages, a greater proportion of people in the UK and elsewhere will be living with dementia, and more of the country's resources will be required to care for them.

### Policy context and relevance
The importance of dementia has been highlighted with the publication of a national dementia strategy for England[39] and NI,[40] and in the ex-Prime Minister Cameron's 'Challenge on Dementia' in 2012.[41] The latter emphasised the central role of three issues: raising awareness of dementia and reducing social isolation that people with dementia feel; improving the quality of care; and increasing research. Good-quality research generates awareness, shapes policy and encourages service development. However, there is an ongoing need for robust estimates of dementia. Evidence based on data from the National Health Service Quality and Outcomes Framework suggests that 56% of dementia cases in England are not formally diagnosed, with wide variations between areas of the country.[42] Therefore, it is hard to reach a clear consensus about the true dementia prevalence in the UK and elsewhere.

### METHODS AND ANALYSIS
### Study aims and objectives
The aim of the NICOLA-HCAP study is to collect detailed cognitive data in a subset (n=1000) of the NICOLA cohort using an enhanced harmonised protocol assessment, with measures that are currently administered in the US-based HRS.[2] This will enable estimation of the prevalence of cognitive impairment and dementia among older adults and comparisons of the rates of dementia prevalence in the USA and Ireland. The TILDA study is conducting their HCAP study concurrently with NICOLA-HCAP, thus enabling the comparison of rates across Ireland. The findings will increase understanding of the personal, social and economic burden of dementia. The study has five main objectives.

### Objective 1: collect in-depth interview assessments
The collection of in-depth interview assessments involves the administration of standardised NICOLA-HCAP interviews, both with respondents and their family member or friend ('informant'). The protocol was developed by the HRS-HCAP for the international network.[43] The respondent's cognitive ability is measured using a CAPI. A family member or friend is asked to complete an additional interview of validated measures to provide information on respondent's prior and current cognitive and physical functioning and whether the individual requires help with daily activities, two core clinical criteria required for dementia diagnosis.[43]

### Objective 2: harmonise scoring and investigate the prevalence of dementia and cognitive impairment
Harmonisation of the scoring of the NICOLA-HCAP data to that of HCAP studies conducted in Ireland (TILDA), the USA (HRS) and England (ELSA) will take place. As the work aims to establish research diagnoses of probable presence or absence of dementia and cognitive impairment, algorithms will be developed to ascertain the probable presence or absence of dementia and cognitive impairment in this subsample of NICOLA, in collaboration with colleagues from TILDA, HRS and ELSA. Establishing clear assessments of dementia or MCI prevalence rates in the study will allow the extrapolation of dementia rates from this subsample to the rest of the NICOLA population in general. As a result, this information will provide the first estimates of dementia and cognitive impairment from a nationally representative sample in NI.

### Objective 3: explore the validity of dementia estimates
The work will involve the construction of estimates of dementia prevalence for the national populations of NI and Ireland harmonised to estimates for the USA and England, having first established the representativeness of the national cohorts, and the development of population weights to account for non-random participation and institutionalisation (care homes). Relationships between the harmonised HCAP data and the more limited cognitive and functioning measures in the core Irish studies will be analysed to establish comparable research diagnoses in the full samples to further expand the data available for epidemiological study, and identify measures that might be added to core studies to improve cognitive assessment going forward.

## Objective 4: investigate the risk factors for dementia and cognitive impairment

Preliminary analyses will be conducted to: (a) investigate potential causal pathways from cardiovascular instability, cerebral microvascular disease and sensory impairment to dementia using objective measures from the TILDA and NICOLA health assessments, including neuroimaging, near-infrared spectroscopy, hearing, vision, retinal photography, and multisensory integration technologies and using polygenic scores to capture genetic differences; (b) use cross-country analyses to investigate how stress and long-term exposure to conflict in a unique cohort of older adults affects cognitive ageing and physiological ageing and methylation/epigenetic patterns in known risk loci related to traumatic stress; and (c) establish potentially modifiable risk factors for primary and secondary prevention of dementia.

## Objective 5: provide reliable data to researchers

Data from NICOLA-HCAP will be provided to the research community to complement existing public data resources from these studies and support epidemiological research being conducted elsewhere.

## Study design

The NICOLA-HCAP assessment comprised two main parts: a face-to-face interview with the participant, and a shorter interview, with a family member or friend, nominated by the participant (the 'informant').

## Participant interview

The participant interview is administered face to face using a CAPI formatted using Epi Info software package.[44] The interview takes place in the participant's home; however, the option of attending the Centre for Public Health at Queen's University Belfast or the Clinical Translational Research and Innovation Centre (C-TRIC) based in Derry/Londonderry is offered to those who prefer not to have a researcher visit them at home. The interview is approximately 1 hour in duration and consists of 19 different measures of memory, language, attention and other aspects of cognitive ability. The Centre for Epidemiological Studies Depression Scale is administered as an assessment of mental well-being to adjust for the impact of depressive symptoms on cognitive test performance (see table 1). The majority of measures are completed verbally by the participant, with the researcher recording answers on the laptop, while some other measures are completed by the participant using paper and pencil (due to nature of the test and licensing restrictions). Interviews with participants are audio recorded (only with participants' consent) on the laptop for internal quality control processes. All participants are offered a gift voucher (£20) in appreciation of their participation in the study.

**Table 1** List of outcome measures from participant CAPI interview[43]

| Outcome measure | Cognitive domain |
| --- | --- |
| Mini-Mental State Examination (MMSE)[48] | Global cognitive status |
| HRS Telephone Interview for Cognitive Status (HRS-TICS) | Global cognitive status |
| CERAD Word List Learning and Recall–Immediate | Episodic memory (immediate) |
| Retrieval Fluency (Animal naming) | Language/fluency |
| Letter Cancellation | Attention/processing speed |
| Backward Count | Attention/processing speed |
| 10/66 Respondent | Global cognitive status |
| CERAD Word List Recall–Delayed | Episodic memory (delayed) |
| Story Recall–Immediate | Episodic memory (immediate) |
| CERAD Word List–Recognition | Episodic memory (recognition) |
| CERAD Constructional Praxis–Immediate | Visuospatial ability |
| Symbol Digit Modalities Test | Attention/processing speed |
| CERAD Constructional Praxis–Delayed | Episodic memory (delayed) |
| Story Recall–Delayed | Episodic memory (delayed) |
| Story Recall–Recognition | Episodic memory (recognition) |
| HRS Number Series | Executive function |
| Raven's Standard Progressive Matrices | Abstract reasoning |
| Trail Making Test (Parts A and B) | Executive function/attention/processing speed |
| CES-D Depressive Symptoms | Assessment of presence and level of depression |

Tests listed in order of administration.
CAPI, computer-assisted personal interview; CERAD, Consortium to Establish a Registry for Alzheimer's Disease; CES-D, Centre for Epidemiological Studies Depression Scale; HRS, Health and Retirement Study.

The NICOLA-HCAP interview is replicated on the HRS study and contains the same measures administered in the USA. These measures are standardised measures for the identification and diagnosis of cognitive impairment and dementia administered across the globe, including in memory clinics in NI. The HCAP assessment was adapted from the Aging, Demographics and Memory Study (ADAMS)[45] which was originally designed to provide the first national estimates of dementia in the USA. The ADAMS study included a 3-hour cognitive assessment. HCAP was designed to be less intensive (ie, 1 hour) than the ADAMS study in order to reduce burden and fatigue for participants. Care is also taken with participants who were physically frail or facing cognitive issues to ensure that they understood the content and purpose of the study, its voluntary nature and that the burden of participating in the study was not too great for them. The research nurses were also experienced in working with older people including those with cognitive decline.

### Family or friend (informant) interview

During the home interview, the participant is asked to nominate a family member or friend (the 'informant') who has known them for some years and whom they think would be willing to answer some questions about the participant and their daily activities. The family or friend interview includes questions relating to changes in the participant's cognitive abilities over time, and activities the participant carries out inside or outside the home (see table 2). As with the participant interview, the family or friend interview is also administered via CAPI in Epi Info[44] and is audio recorded for quality control purposes. The interview lasts approximately 20 min and is administered

**Table 2** List of outcome measures from informant CAPI interview[43]

| Outcome measure | Description |
| --- | --- |
| Family or friend demographics | Demographic profile of informant |
| Jorm IQCODE | Assessment of cognitive decline |
| Blessed Dementia Rating Scale–Part 2 | Assessment of ability to do basic self-care activities |
| HRS Activities Questionnaire | Assessment of participant activity engagement |
| CSI-D Cognitive Activities Questionnaire | Assessment of participant's cognitive activity engagement and ability |
| 10/66 Informant Questionnaire | Assessment of ability to do daily activities |
| Blessed Dementia Rating Scale–Part 1 | Assessment of additional activities and mental ability |

Tests listed in order of administration.
CAPI, computer-assisted personal interview; CSI-D, Community Screening Instrument for Dementia; HRS, Health and Retirement Study; IQCODE, Informant Questionnaire on Cognitive Decline in the Elderly.

at the participant's home if the family member or friend is present at the time of the participant interview. If they are not present, the family member or friend is contacted by telephone and the interview is completed over the phone at a time convenient to them. Alternatively, the family member or friend can request a paper version of the questionnaire to be posted to their home or they can complete the questionnaire online (based on Qualtrics user platform[46]), whereby a link to the survey is emailed or texted to their mobile phone. The participant can request someone to be present during their interview for support; however, it is preferred the informant interview is completed one to one with the research nurse (or online) to minimise bias based on the presence of the participant.

### Sample

The NICOLA-HCAP sample is drawn from core members of the NICOLA cohort. The aim is to recruit 1000 participants who took part in the NICOLA Wave 2 CAPI (end date December 2019), aged 65 years or over at the start of HCAP fieldwork. The data from this sample are being combined with the same HCAP data from our sister study TILDA based in Ireland and the ELSA study in England. The combined sample is equivalent to the sample size of the HRS-HCAP sample.

Both participant and informant interviews are conducted with participants who are residing in private residential accommodation in NI. Those who were in institutionalised care, that is, residential homes or nursing homes, were excluded from participating in NICOLA and therefore will not be included in NICOLA-HCAP. Individuals are randomly selected and the sample is stratified to include participants who live alone (single household) and participants who do not live alone (multiple household). Participants in multiple households who are initially ineligible but become eligible later (typically spouses who turn 65 after the onset of the study) are also able to take part. An 80% response rate is expected to the NICOLA-HCAP, which implies a selection rate of about 1 in 3 to achieve the target of 1000 participants. Individuals will be excluded from the study if they are unable to complete the written tasks included in the NICOLA-HCAP assessment (due to reasons such as sensory impairment; eg, hearing loss, visual impairment) or if they are not fluent in English. As the proportion of non-English speakers was low in the NICOLA over 65 cohort, the HCAP assessment was not translated to other languages. However, other studies in the network have translated the assessment and therefore it is possible for further NICOLA-HCAP waves.

A protocol was developed for those individuals who were deemed to not have capacity to consent. In the main waves of NICOLA, if a participant is thought to not be mentally capable of giving informed consent (if, for example, they are too confused to be able to fully understand what they are consenting to), a consultee can be approached to decide whether or not the participant should be included

in that wave of NICOLA. For the HCAP substudy, if it is deemed that the participant does not have the capacity to consent, they can nominate a friend/relative or carer to act as a consultee. The consultee will be asked by the researcher whether they believed that, were the participant able to consent, they would agree to take part in the HCAP substudy within NICOLA. The consultee will then sign a Consultee Declaration Form confirming that in their opinion the participant would or would not want to take part in the study.

Informants are nominated by participants and can be a family member, friend or carer. The researcher discusses the nomination with the participant once they have agreed to take part in the interview, and recommends that the nominated person is someone who knows the participant well and sees them frequently enough to answer questions about their daily life, as well as being someone who has known them for a number of years. The recommendation is that the nominated person is a family member or friend rather than someone who is employed to be with the participant, such as a carer; however, if a family member or friend is not an option and the employed person has known the participant for long enough, they can fulfil this role. Inability to nominate an informant is not an exclusion criterion; should the participant refuse to nominate or the informant refuse to take part, NICOLA-HCAP will follow the HRS protocol for the imputation of missing data.

### Proposed statistical methods

We will develop algorithms to ascertain the probable presence or absence of dementia and cognitive impairment in this subsample of NICOLA, in collaboration with colleagues from TILDA, HRS and the ELSA. Establishing clear estimates of dementia or MCI prevalence rates in the study will allow the extrapolation of dementia rates from this subsample to the rest of the NICOLA population in general. As a result, we will be able to provide the first estimates of dementia and cognitive impairment from a nationally representative sample in NI.

We will harmonise the scoring of the Northern Irish HCAP data to that of HCAP studies conducted in Ireland (TILDA), the USA (HRS) and England (ELSA), and establish research diagnoses of probable presence or absence of dementia and cognitive impairment in these subsamples of NICOLA using algorithms developed in collaboration with HRS and ELSA based on harmonised measures. We will construct estimates of dementia prevalence for the national populations of Ireland and NI harmonised to estimates for USA and England, having first established the representativeness of the national cohorts, and develop population weights to account for non-random participation and institutionalisation (care homes). Sample weights will be applied as per the HRS protocol adjusting for differential sampling weights by race/ethnicity, age, cognitive ability and for differential participation by demographic factors such as education, socioeconomic status, etc.

The use of the 1-hour battery of identical tests, 20 min of informant interview and similar methods of administration provides a strong foundation for harmonisation. Subtle differences in scoring can produce disparities, so there will be extensive comparison across studies using audio recordings, participant drawings and other records to ensure comparability. Led by HRS, statistical procedures such as tests for factor invariance will be used to determine if there are any specific inconsistencies in tests between countries. Similar analyses will be carried out with the informant reports.

We will analyse the relationships between the harmonised HCAP data and the more limited cognitive and functioning measures in the core Irish studies to establish comparable research diagnoses in the full samples to further expand the data available for epidemiological study, and identify measures that might be added to core studies to improve cognitive assessment going forward.

We will conduct preliminary analyses to: (a) investigate potential causal pathways from cardiovascular instability, cerebral microvascular disease and sensory impairment to dementia using objective measures in TILDA and NICOLA health assessments, including neuroimaging, near-infrared spectroscopy, hearing, vision, retinal photography, and multisensory integration technologies and using polygenic scores to capture genetic differences; (b) use cross-country analyses to investigate how stress and long-term exposure to conflict in a unique cohort of older adults affects cognitive ageing and physiological ageing and methylation/epigenetic patterns in known risk loci related to traumatic stress; and (c) establish potentially modifiable risk factors for primary and secondary prevention of dementia.

### Staff training

Fieldwork staff working on the NICOLA-HCAP study includes both Queen's University research staff and dementia-trained clinical research nurses from the Northern Ireland Clinical Research Network. All staff received face-to-face training delivered in two phases. Phase 1 focused on study orientation and the development of the standard operating procedures (SOP). In brief, sessions with fieldwork staff included an introduction to the research aims and objectives; study design; staffing roles; recruitment procedures and previsit protocols; consent procedures and capacity to consent protocols; set-up of Epi Info software[44] (for data capture—participant CAPI and informant CAPI) and the upload of data to secure servers. Phase 2 focused on the administration of cognitive tests and informant measures. The Survey Research Operations (SRO) team based in the Survey Research Center in the University of Michigan developed training videos and scripts for in-house training. Fieldwork staff met in the first instance as a full team to review the videos and practice administration using the CAPI software. This was followed by practice administration using a buddy system. This involved fieldwork staff working in pairs, with one member of staff adopting the

role of the participant and the other the interviewer (and vice versa). As training for TILDA-HCAP was running consecutively with NICOLA-HCAP, joint TILDA-NICOLA meetings were held with the SRO and HRS team to address any queries relating to test administration and scoring. The HRS team subsequently developed a scoring protocol to ensure consistency across studies.

### Patient and public involvement

The pilot study included participants from the NICOLA Healthy Ageing Research Advisory Group. This group consisted of both NICOLA participants and individuals from AgeNI, a local NI-based charitable organisation. These participants gave feedback during the pilot phase in relation to the length of the assessment, content and overall process. Further detail is noted below.

### NICOLA-HCAP dress rehearsal

The pilot dress rehearsal mirrored the ELSA-HCAP dress rehearsal[47] and was used to inform the mainstage survey. The main aim of the pilot was to implement the study as per the NICOLA-HCAP study protocol and to identify any areas that needed to be revised or improved in relation to the study design, questionnaire format and content, data collection, data upload processes and materials prior to the main phase and the SOP could be amended accordingly. The pilot dress rehearsal was conducted between 6 January 2022 and 25 January 2022.

### Pilot design

The dress rehearsal aimed to recruit 15 participants and 15 family or friend members to take part in the respective interviews. Following each interview, the research nurses respectively completed an interview feedback form that included sections for feedback on both fieldwork procedures and in-house processes. In brief, the fieldwork feedback included information on personal protective equipment requirements and any perceived impact on study administration; participant information and consent processes; CAPI and paper administrations; and participant experience. Feedback on the in-house processes centred on the administrative procedures; previsit protocols; use of secure servers and the efficiency of the process required to carry out data uploads of participant and family or friend interviews; data storage and any additionally required processes to support the smooth delivery of the project.

### Pilot sample

A total of 14 participant interviews and 14 informant interviews were achieved within the timeframe for the pilot study. The pilot dress rehearsal sample was selected from the NICOLA Healthy Ageing Research Advisory Panel which included both NICOLA participants (n=5) and volunteers from AgeNI[48] (n=9) along with their respective family member or friend. Members of the Advisory Panel understood that their role was to test the design, content and materials of the proposed NICOLA-HCAP interview. Ethical approval for the pilot was obtained from the Faculty of Medicine, Health and Life Sciences Research Ethics Committee, Queen's University Belfast.

### Pilot revisions

Feedback obtained from fieldwork staff informed a number of study revisions or improvements that were integrated into the SOP and implemented for the main study. The pilot identified the following key areas for improvement.

*Area 1: Previsit documentation.* Fieldwork staff reflected on the complexities of test administration and the transition between the CAPI and paper versions of tests. Previsit protocols were revised to include preparation processes (ie, folders with test papers organised in order of test administration and participant identification numbers preadded to documentation). Staff also preferred a manual timer for tests rather than the built-in CAPI version.

*Area 2: Test administration.* Queries were raised in relation to the logistics of some test administration. Fieldwork staff also questioned how tests were scored and entered into the Epi Info software. All queries were resolved with the HRS and TILDA teams.

*Area 3: Study administration processes.* Additional administration databases were set up to ease the flow of recruitment and home visit bookings. In addition, the SOP was amended to update in-house filing and the secure storage of data.

### ETHICS AND DISSEMINATION

The main study received ethical approval from the Faculty of Medicine, Health and Life Sciences Research Ethics Committee, Queen's University Belfast.[49] Participants and informants provided written informed consent prior to participation in the study. The NICOLA-HCAP main study commenced on 7 February 2022 with fieldwork due to complete in June 2023. The data from NICOLA-HCAP are currently held in the data repository within the Centre for Public Health at Queen's University Belfast and after study completion, all NICOLA-HCAP data will be provided to the research community via data repositories such as the Dementias Platform UK[50] and Gateway to Global Aging to complement existing public data resources and support epidemiological research by others. NICOLA Wave 1 and Wave 2 data are also maintained and stored at a data repository within the Centre for Public Health, Queen's University Belfast. Information on how to access NICOLA data and details regarding the data that are currently available in NICOLA can be found on the study website.[51] Findings from this study will also be disseminated through peer-reviewed publications and via presentations at international conferences.

**Acknowledgements** The authors acknowledge the participation of members from the NICOLA Healthy Ageing Research Advisory Panel in the NICOLA-HCAP pilot study. Their participation and feedback on the process was invaluable to the study. The authors are also grateful to the participants from the NICOLA study and

the wider NICOLA team for their continued support and assistance, and thank the research nurses, clerical staff and computer technicians for their time, dedication, professionalism and contribution to the NICOLA-HCAP study.

**Contributors** LO'H, CN, CM, MM, FK, DW and BM have made substantial contributions to the conception or design of the work; or the acquisition, analysis or interpretation of data for the work. LO'H, CN, CM, MM, FK, DW and BM were involved in drafting the work or revising it critically for important intellectual content. LO'H, CN, CM, MM, FK, DW and BM gave final approval of the version to be published and agreed to be accountable for all aspects of the work in ensuring that questions related to the accuracy or integrity of any part of the work are appropriately investigated and resolved.

**Funding** The Northern Ireland Cohort for the Longitudinal Study of Ageing Harmonised Cognitive Assessment Protocol (NICOLA-HCAP) is funded by the National Institute on Aging (NIA; grant number: SUBK00010180). The NICOLA study (Waves 1 and 2) is funded by the Atlantic Philanthropies (grant number: 18107); the Economic and Social Research Council (grant number: ES/L0084559/1); the UKCRC Centre for Excellence for Public Health Northern Ireland; Northern Ireland Health and Social Care Research and Development Division of the Public Health Agency (grant number: STL/4717/12); the Wellcome Trust/Wolfson Foundation; the Centre for Ageing Research and Development Ireland; Queen's University Belfast; and the Office of the First Minister and Deputy First Minister Northern Ireland.

**Disclaimer** The authors alone are responsible for the interpretation of the data and any views or opinions presented are solely those of the authors and do not necessarily represent those of the NICOLA study team.

**Competing interests** None declared.

**Patient and public involvement** Patients and/or the public were involved in the design, or conduct, or reporting, or dissemination plans of this research. Refer to the Methods section for further details.

**Patient consent for publication** Obtained.

**Provenance and peer review** Not commissioned; externally peer reviewed.

**ORCID iD**
Leeanne O'Hara http://orcid.org/0000-0001-7735-8148

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
