## [Reviewer comments · BMJ Open]

ARTICLE DETAILS

TITLE (PROVISIONAL)	Investigating the prevalence of Cognitive Impairment and Dementia in the Northern Ireland Cohort for the Longitudinal Study of Ageing (NICOLA): The Harmonised Cognitive Assessment Protocol (HCAP) Cross-sectional Sub-study.
AUTHORS	O'Hara, Leeanne; Neville, Charlotte; Marr, Calum; McAlinden, Michael; Kee, Frank; Weir, David; McGuinness, Bernadette

VERSION 1 – REVIEW

REVIEWER	Barbora Silarova University of Kent, Personal Social Services Research Unit
REVIEW RETURNED	06-Nov-2023

GENERAL COMMENTS	Title and abstract: the study design is clearly stated in the title and abstract. The abstract is an informative summary of what will be done. Background clearly describes the scientific background and rationale for this study. The objectives of the study are clearly stated. The key elements of the study design are clearly defined. It would be good to get some additional clarification regarding following points: • Could the authors comment on whether the participant interview has been piloted e.g. with people with dementia/MCI? They do describe it was indeed piloted but I was unclear whether those people had dementia or MCI? If yes, could they comment on any changes to their protocol as a result of the pilot testing with people with dementia/MCI? E.g. one hour interview for people living with dementia/MCI can be incredibly taxing – is there a possibility to break this down, e.g. to two visits – I can understand this may be not possible but would be good if the research team share their reflections how they navigated the decisions around participant's burden, following protocols from other studies etc.....• Was there any PPI involvement in this study?• It would be good to understand whether measures listed in Table 1 and Table 2 were validated and tested in Northern Ireland. If not it would be good to discuss this in the Limitations section.• I assume that the participant and their informant will be interviewed/tested separately. Is this correct, what if the people want to be present during their interviews/testing, is that an option?• It would be good if authors states explicitly inclusion/exclusion criteria. E.g. I was unclear whether only people who can consent can take part. I assume anyone should be able to take part as the aim is to calculate the estimates of dementia/MCI. Importantly you listed some exclusion criteria e.g. Individuals will be excluded from the study if they are unable to complete the written tasks included in the NICOLA-HCAP assessment (due to reasons such as sensory impairment, e.g. hearing loss, visual impairment) or if they are not fluent in English – can you provide an explanation how you are
--

	going to deal with this, as e.g. people with dementia/MCI experience these issues. It would be good to also reflect on exclusion of those whose English is not a native language.  • I understand from your protocol the focus is on people who have family member/friend willing to participate. Again it would be good to reflect on those who may not have an informant and that their voices/experiences are repeatedly excluded from similar research. What is the impact of this on achieving your objectives: e.g. objective 3 and 4..... • It would be good to understand how you calculated the sample size for this study • Could you add a list of all outcomes for this study? • Would it be possible to add more details regarding proposed statistical methods?
--	--

VERSION 1 – AUTHOR RESPONSE

REVIEWER 1	
Title and abstract: the study design is clearly stated in the title and abstract. The abstract is an informative summary of what will be done. Background clearly describes the scientific background and rationale for this study. The objectives of the study are clearly stated.	Thank you for your comments.
The key elements of the study design are clearly defined. It would be good to get some additional clarification regarding following points:  • Could the authors comment on whether the participant interview has been piloted e.g. with people with dementia/MCI? They do describe it was indeed piloted but I was unclear whether those people had dementia or MCI? If yes, could they comment on any changes to their protocol as a result of the pilot testing with people with dementia/MCI? E.g. one hour interview for people living with dementia/MCI can be incredibly taxing – is there a possibility to break this down, e.g. to two visits – I can understand this may be not possible but would be good if the research team share their reflections how they navigated the decisions around participant’s burden, following protocols from other studies etc..... 	Now added to the study design. See page 9.
 • Was there any PPI involvement in this study? 	Now added to the study design. See page 13.
 • It would be good to understand whether measures listed in Table 1 and Table 2 were validated and tested in Northern Ireland. If not it 	Now added to study design.

would be good to discuss this in the Limitations section.	See page 9.
• I assume that the participant and their informant will be interviewed/tested separately. Is this correct, what if the people want to be present during their interviews/testing, is that an option?	Now added to the family or friend interview section. See page 10.
• It would be good if authors states explicitly inclusion/exclusion criteria. E.g. I was unclear whether only people who can consent can take part. I assume anyone should be able to take part as the aim is to calculate the estimates of dementia/MCI. Importantly you listed some exclusion criteria e.g. Individuals will be excluded from the study if they are unable to complete the written tasks included in the NICOLA-HCAP assessment (due to reasons such as sensory impairment, e.g. hearing loss, visual impairment) or if they are not fluent in English – can you provide an explanation how you are going to deal with this, as e.g. people with dementia/MCI experience these issues. It would be good to also reflect on exclusion of those whose English is not a native language.	Now added to the sample section. See page 11.
• I understand from your protocol the focus is on people who have family member/friend willing to participate. Again it would be good to reflect on those who may not have an informant and that their voices/experiences are repeatedly excluded from similar research. What is the impact of this on achieving your objectives: e.g. objective 3 and 4.....	Now added. See page 12.
• It would be good to understand how you calculated the sample size for this study	Now added to the sample section. See page 10.
• Could you add a list of all outcomes for this study?	Added to tables 1 and 2.
• Would it be possible to add more details regarding proposed statistical methods?	Section on proposed statistical methods has been revised. See page 12.

VERSION 2 – REVIEW

REVIEWER	Barbora Silarova University of Kent, Personal Social Services Research Unit
REVIEW RETURNED	04-Jan-2024
GENERAL COMMENTS	All the queries have been addressed. Thank you.